# Studying of Cultural Properties of *Pyricularia oryzae* Cav. Strains in the South of Russia

**Dmitry Nartymov [1], Evgeny Kharitonov [1], Elena Dubina [1,*], Sergey Garkusha [1], Margarita Ruban [1], Nikita Istomin [1] and Pavel Kostylev [2]**

[1]    Federal Scientific Rice Centre, Belozerny, 3, 350921 Krasnodar, Russia; dimnortey@mail.ru (D.N.); arrri_kub@mail.ru (E.K.); l.esaulova@mail.ru (S.G.); rubanmarg@gmail.com (M.R.); istomin_nike@mail.ru (N.I.)

[2]    Agrarian Research Center "Donskoy", Nauchny Gorodok, 3, 347740 Zernograd, Russia; p-kostylev@mail.ru

[*]    Correspondence: lenakrug1@rambler.ru; Tel.: +7-918-432-65-82

**Abstract:** This article presents the results of the development of a methodology for describing the main morphological and cultural traits of the *Pyricularia oryzae* Cav. strains widespread in the south of Russia. At the same time, the types of traits are identified and listed, which make it possible to unambiguously determine the uniqueness and variety of the pathogen. The relationships and patterns established using cluster and statistical analysis make it possible to identify the conditions for the development of a pathogen that determine its predominant forms. Thus, research shows that leaf forms of *P. oryzae* strains isolated from rice plants with leaf form of blast disease have an equally directional growth pattern of a colony with a felt structure, and strains isolated from neck-affected plant form often produce a zone of a colony with a clumpy structure. The classification of cultural traits will make it possible to obtain scientifically grounded and comparable data that can be used in the analysis of the interaction of *P. oryzae* strains with rice plants on various varieties and in various agro-technological conditions in order to improve and rationalize agricultural activities. The study opens up the possibility of using data in breeding, making it possible to identify forms of a pathogen that infect certain varieties.

**Keywords:** rice blast; morphological and cultural traits; strains

## 1. Introduction

Within the framework of research carried out in the field of studying the cultural and morphological traits of the *Pyricularia oryzae* Cav. strains, it is often necessary to classify cultural characteristics. This is due to the fact that when identifying traits, as a rule, a subjective approach is used, determined directly by the researcher, which significantly complicates the comparability of the results of various studies when studying rice blast strains. In addition, the absence of any method for describing cultural properties hinders the development of a scientifically grounded basis for identifying patterns of pathogen development in terms of morphological and cultural characteristics.

Analysis of literary and scientific sources shows that the identification of cultural traits of strains of microscopic fungi, similar to rice blast, is reduced to a descriptive character and, as a rule, does not have a clearly expressed typification according to any features of the colony development [1–6]. Thus, while one of the methods involves the identification of the structure, surface, color, shape, and edge of the colony, the others are based on the determination of texture, growth rate of the colony, color of aerial and substrate mycelium [7–9]. Some techniques involve the detection of exudate and the presence of pigment released into the environment [10,11].

The disunity of the methods is largely determined by their applicability to specific types of strains of phytopathogens and microorganisms, since, although they are similar to the types of microscopic fungi, they differ in many ways in the manifestation of certain

characteristics [12]. At the same time, general scientifically grounded approaches to the description of the cultural properties of a colony of *P. oryzae* strains are not presented in the scientific literature, although attempts to identify general patterns of colony development are encountered [13], and thus on the basis of the research work of the laboratory of information digital and biotechnologies of the Federal State Budgetary Scientific Institution Federal Scientific Rice Centre, we collected a large amount of material on the study of morphological and cultural traits and genotyping of *P. oryzae* strains common in southern Russia [13–18].

## 2. Materials and Methods

The research work was carried out in two interrelated stages. At the first stage, the identification and classification of cultural traits of the *P. oryzae* strains was carried out. The second stage of the work involved the clustering of the studied strains according to the identified characteristics in order to determine the patterns and relationships in the manifestation of the properties and conditions of the pathogen development in agricultural production.

To identify and classify the main cultural traits that characterize the colony of rice blast strain, we selected data based on the results of studies carried out in 2015–2020, which we systematized in the database [17]. The data used had a rather fragmented format for describing cultural properties, which did not define the set of traits unambiguously, making it difficult to classify and compare strains of the *P. oryzae* pathogen.

The data were subjected to statistical and analytical processing for grouping and identifying characteristic points indicating the presence of a detectable trait. Further, the characteristic points were associated with a trait, and the data of a particular strain were associated with a set of inherent classified traits. Thus, it became possible to carry out a comparative analysis of the studied strains of rice blast, and the description of the cultural traits received a clear classification.

At the second stage of the study, a general hierarchical clustering was carried out according to the classification cultural characteristics obtained at the first stage.

Cluster analysis was carried out using the Ward method based on applying standardized data on the presence of a trait species [19]. The presence of a type of trait was determined by the maximum standardized value of 1, and the absence-0. The distance between the clustering points was determined by a set of standardized values for each strain.

Clustering data were statistically analyzed to identify patterns characterizing the resulting clusters, according to cultural traits and other characteristics of *P. oryzae* strains.

## 3. Results

Within the framework of the classification carried out, we identified six main cultural trait characteristics of the strains under study: the nature of the colony growth, the developed colony structure, the relief of the colony surface, the developed colony profile, the color of the aerial mycelium, and the color of the substrate mycelium. The nature of the colony growth determines the direction and characteristics of the colony development in the space above the nutrient medium, as a rule, in the direction from the center of the Petri dish to its edges. Thus, five variants of the manifestation of this trait were identified: equidirectional, zoned, concentric, radiant, and multidirectional (Figure 1).

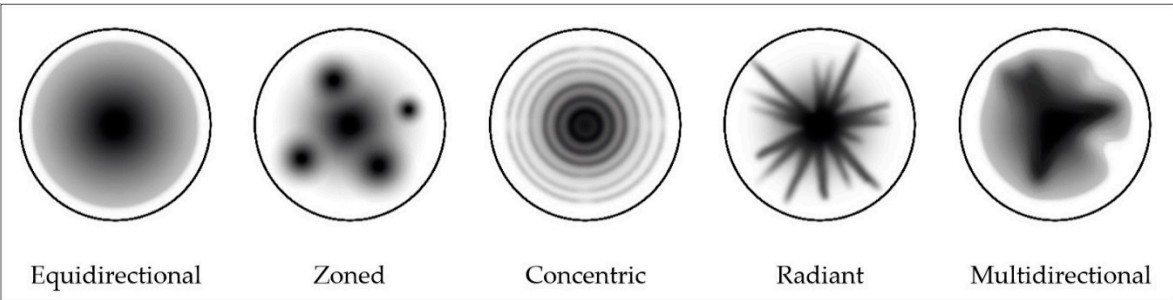

**Figure 1.** Nature of the colony growth of *Pyricularia oryzae* Cav. strain.

The equidirectional nature of the colony growth is determined by the uniform development of the colony in all directions from the center to the edges of the Petri dish and forms, as a rule, a rounded colony inclusion in the plane of growth. This type of colony growth is determined by the sequence of mycelium growth during the entire observation period.

The zoned nature is determined by uneven or sectoral colony growth in the form of separate growth. In this case, equidirectional mycelium growth can be observed within one zone. In general, the intensity of the development of the colony is not the same, and the nature of the location of the zones is not defined.

The concentric nature of growth indicates the presence of a wavy development of the colony, directed from the center of the colony to its edges, which manifests itself in the form of concentric rings in the plane of growth of the colony. This is largely due to the heterogeneous, undulating growth rate of the colony over time. In this case, the unevenness spreads in the same direction.

The radiant nature of growth is determined by the presence of pronounced rays of intensive colony development towards the edges of the colony. The multidirectional nature of growth is manifested in the alternation of the intensity of the colony development in different directions.

The developed colony structure describes the features of the structural formation of the colony in the thickness of the aerial mycelium. Analysis of this trait showed that the colonies of the studied strains took five different forms: homogeneous, fluffy, felt, powdery, and clumpy (Figure 2).

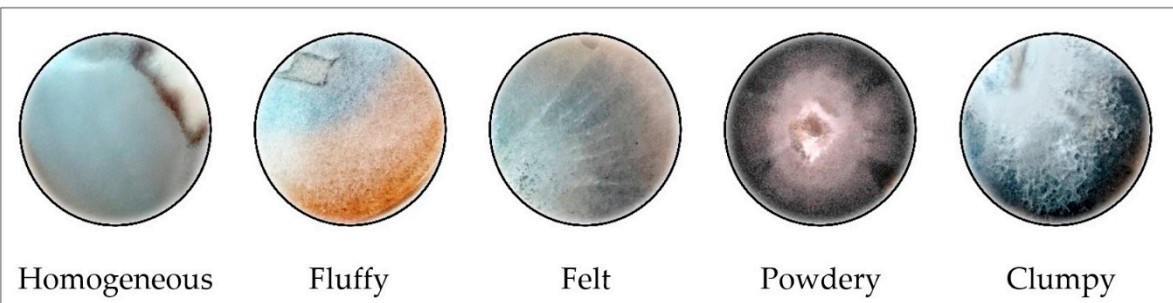

**Figure 2.** Colony structure of *Pyricularia oryzae* Cav. strain.

The homogeneous structure of a developed colony is characterized by a uniform structure of the mycelium without friability and gaps. Uniformity is manifested throughout the mycelium tissue and has a fairly high density of mutual arrangement of filaments.

The fluffy structure is characterized by a noticeable fluff-like mycelium tissue heterogeneity. However, in general, the structure is uniform, but the density of mycelium filaments is noticeably lower than with a homogeneous structure.

The felt structure is characterized by the presence of pronounced fiber-like compaction. It is characterized by the presence of denser interweaving of mycelium filaments with small gaps.

The powdery structure is characterized by the presence of small nodes of a denser accumulation of mycelial bodies located on the loose tissue of the mycelium.

The clumpy structure of the mycelium is determined by the presence of pronounced seals, alternating with less dense areas of mycelium tissue.

The relief of the colony surface determines the unevenness of the surface formed during the colony development. In the studied strains, four types of relief were identified: conical, convex, bumpy, and uniform.

The conical relief of the colony has a crater-like appearance, in which the height of the colony development decreases exponentially from the center to the edges of the colony. The convex relief is characterized by a drop-like shape. Bumpy relief describes the presence of both high and low smoothly transitioning areas. Uniform relief characterizes a colony evenly distributed over the entire surface of the substrate (Figure 3).

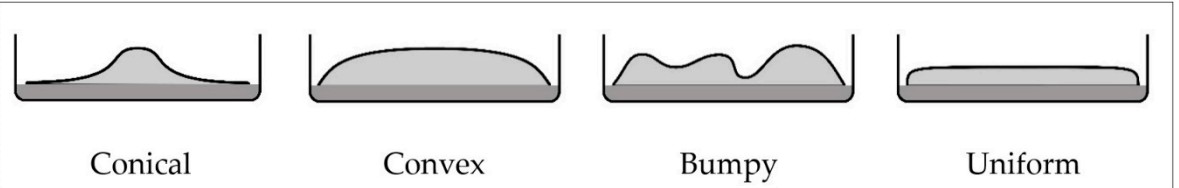

**Figure 3.** Relief of the colony surface of *Pyricularia oryzae* Cav. strain.

The colony profile determines the height of the mycelium surface from the substrate surface relative to the average line: high, medium, and low.

Aerial mycelium color of *P. oryzae* strains is predominantly gray in color, and therefore we distinguished them as light gray, typical gray, dark gray, olive gray, and beige gray. From the colors of the substrate mycelium, we distinguished discolored, gray, black, brown, and brown strains. The main cultural traits are described in Table 1.

**Table 1.** Characterization of clusters according to the cultural traits of a colony of *Pyricularia oryzae* Cav. strains.

|    | Nature of the Colony Growth | Colony Structure | Surface Relief | Colony Profile | Aerial Mycelium Color | Substrate Mycelium Color |
|----|------------------------------|------------------|----------------|----------------|------------------------|---------------------------|
| A1 | Multidirectional | Powdery | Bumpy   | Medium | Beige gray   | Black |
| A2 | Zoned            | Powdery | Bumpy   | Low    | Olive gray   | Black |
| A3 | Zoned            | Clumpy  | Bumpy   | High   | Olive gray   | Brown |
| A4 | Zoned            | Clumpy  | Bumpy   | Medium | Beige gray   | Brown |
| A5 | Zoned            | Clumpy  | Bumpy   | Low    | Dark gray    | Black |
| B1 | Radiant          | Powdery | Conical | Low    | Beige gray   | Black |
| B2 | Equidirectional  | Powdery | Conical | Low    | Light gray   | Black |
| B3 | Equidirectional  | Clumpy  | Conical | Low    | Beige gray   | Brown |
| B4 | Multidirectional | Powdery | Conical | Low    | Light gray   | Brown |
| B5 | Equidirectional  | Powdery | Conical | Low    | Typical gray | Gray  |
| C1 | Equidirectional  | Felt    | Conical | Low    | Olive gray   | Brown |
| C2 | Equidirectional  | Felt    | Conical | Low    | Typical gray | Brown |
| C3 | Equidirectional  | Felt    | Uniform | Medium | Olive gray   | Black |
| C4 | Equidirectional  | Felt    | Uniform | Low    | Olive gray   | Brown |
| C5 | Equidirectional  | Fluffy  | Uniform | Low    | Typical gray | Black |

After the classification was carried out, the characteristic points of the presence of the identified traits were established and the links were established for each strain of *P. oryzae* with a certain variant of each of the six traits. Thus, each of the studied strains received a complete systematized description of the cultural characteristics of the colony it forms.

To identify hierarchical relationships of *P. oryzae* strains by cultural traits, we carried out a cluster analysis of cultural traits. This approach made it possible to reveal the presence of classification properties of the typing obtained at the previous stage. Clustering was carried out according to the Ward method with standardization for the presence or absence of a cultural trait. The dendrogram obtained as a result of the cluster analysis is shown in Figure 4.

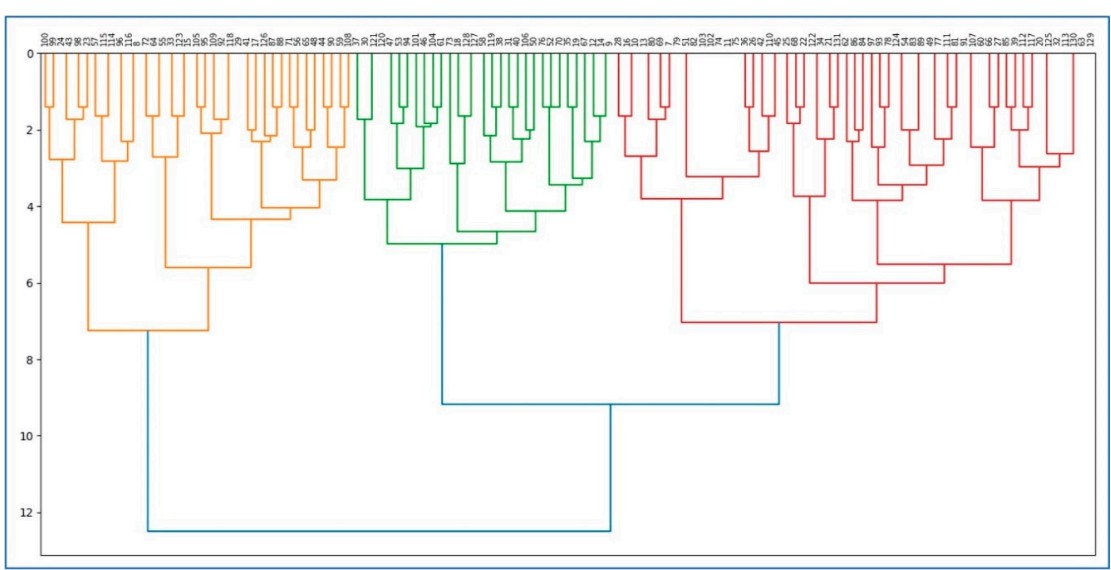

**Figure 4.** Dendrogram obtained as a result of cluster analysis.

It can be seen from the dendrogram that three large clusters can be distinguished, each of which contained five subclusters. Large clusters were usually denoted by the Latin letters A, B, and C, and subclusters were denoted by numbers. Thus, the identified clusters revealed 15 types of *P. oryzae* strains with different cultural traits, forming hierarchical relationships that characterize the commonality of the strains.

The distribution of cluster strains by the sampling site is shown in Figure 5.

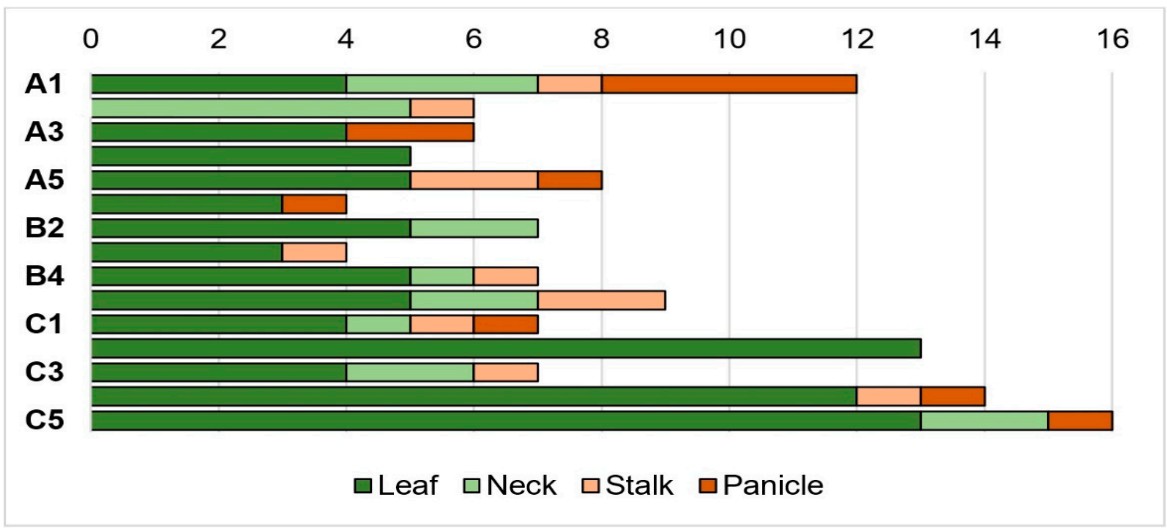

**Figure 5.** Diagram of the distribution of the selection site for rice blast strains by clusters.

The analysis of the interrelation of the clustering data with the geography of selection and the varietal composition of the affected rice plants did not reveal any regularities, which indicates the widespread prevalence of various types of pathogens throughout the south of Russia and the varying degree of instability of all studied varieties. However, clusters B1, B3, and C1 are characterized by the identification of strains selected on plants of the variety Flagman, which can determine the high instability of this variety to blast strains belonging to these clusters.

Analysis of cluster data at the place of selection of strains showed that clusters of group A are characterized by all forms of blast disease: leaf, neck, and panicle. On the other hand, for groups B and C, the most typical is leaf form, and for group C, leaf forms make up about 90%. At the same time, the A2 cluster with a predominance of the neck form; C2, consisting of leaf form strains; and C4 and C5, with a predominance of the leaf form, are clearly distinguished.

## 4. Discussion

As a result of the work carried out, we developed a method for describing the main cultural traits of the *P. oryzae* strains widespread in the south of Russia. These traits included the nature of the colony growth, the developed colony structure, the relief of the colony surface, the developed colony profile, the color of the aerial mycelium, and the color of the substrate mycelium. At the same time, the types of traits were identified and listed, which make it possible to unambiguously determine the uniqueness and variety of the pathogen.

In addition, the relationships and patterns established using cluster and statistical analysis made it possible to identify the conditions for the development of a pathogen that determines its predominant forms. Thus, the research showed that leaf forms of *P. oryzae* strains isolated from rice plants with leaf form of blast disease had an equally directional growth pattern of a colony with a felt structure, and strains isolated from neck-affected plant form often produced a zone of a colony with a clumpy structure. Moreover, such an analysis made it possible to identify the relationship of a group of strains capable of infecting rice plants belonging to the variety Flagman.

## 5. Conclusions

The classification of cultural traits will allow for the obtaining of scientifically grounded and comparable data that can be used in the analysis of the interaction of blast strains with rice plants on various varieties and in various agro-technological conditions in order to improve agricultural activities. In addition, this study opens up the possibility of using data in breeding, making it possible to identify forms of the pathogen that infect certain varieties.

**Author Contributions:** Conceptualization, D.N. and E.D.; methodology, D.N. and M.R.; software, N.I.; validation, D.N., M.R., and P.K.; formal analysis, D.N.; investigation, D.N., M.R., and E.D.; resources, S.G.; data curation, E.K. and E.D.; writing—original draft preparation, D.N. and E.D.; writing—review and editing, D.N., E.D., and N.I.; visualization, D.N.; project administration, E.K. and S.G.; funding acquisition, E.D. All authors have read and agreed to the published version of the manuscript.

**Funding:** The research was carried out with the financial support of the Kuban Science Foundation in the framework of the scientific project no. 20.1/3 "A study of the genetic diversity of rice blast *Pyricularia oryzae* Cav. agent using a complex of molecular and morphological traits to justify the methods of phytosanitary and environmental stabilization of agrophytocenoses in the south of Russia".

**Institutional Review Board Statement:** The study was approved on the Scientific Board meeting of the Federal Scientific Rice Centre (protocol code-5 and date of approval—11 May 2020).

**Informed Consent Statement:** Informed consent was obtained from all subjects involved in the study.

**Acknowledgments:** The Kuban Science Foundation and Federal Scientific Rice Center.

**Conflicts of Interest:** The authors declare no conflict of interest.

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
