# Peer review of "Studying of Cultural Properties of Pyricularia oryzae Cav. Strains in the South of Russia"

_2036-7481, doi:10.3390/microbiolres12010003_

Round 1

Reviewer 1 Report

I have reviewed the MS and found it acceptable with minor revisions. It is well presented although not particularly innovoative. However it does provide useful information to researches on rice blast which is the most important rice diesease in many countries. Here are the revisions suggested. (1) Pyricularia oryzeae must be in itlilics throughout the entire MS. (2) line 21 change studied to study (3) line 22 delete approaches, change open to opens (4) line 101 delete tissue (5) line 103 change mycelial bodies to mycelia; loose tissue to less dense area; mycelium to colony (6) line 105 change seals to nodes; mycelial tissue to colony (7) line 143 change instability to one degree or another to varying degree of instability (8) Discussion: to be consistent use either all present or past tense. Could be expanded by comparing the significance or present findings to the previous literature. (9) Conclusion: line 168 delete rationalize; line 169 studied approaches to study (9) Conclusions: line 168 delete rationalize; studies approaches to study

Author Response

Point 1: Pyricularia oryzae must be in italics throughout the entire MS.

Response 1: We have changed the style of printing type throughout the whole manuscript.

Point 2: line 21 change studied to study.

 Response 2: We have changed the structure of the sentence according to reviewer’s notices.

Point 3: line 22 delete approaches, change open to opens.

 Response 3: We have changed the structure of the sentence according to reviewer’s notices.

 Point 4: line 101 delete tissue.

 Response 4: We have deleted “tissue”.

 Point 5: line 103 change mycelial bodies to mycelia; loose tissue to less dense area; mycelium to colony.

 Response 5: We have changed mycelial bodies to mycelia; loose tissue to less dense area; mycelium to colony.

 Point 6: line 105 change seals to nodes; mycelial tissue to colony.

 Response 6: We have changed seals to nodes and mycelial tissue to colony.

 Point 7: line 143 change instability to one degree or another to varying degree of instability.

 Response 7: We have changed instability to one degree or another to varying degree of instability.

 Point 8: Discussion: to be consistent use either all present or past tense. Could be expanded by comparing the significance or present findings to the previous literature.

 Response 8: We have corrected the tenses of the verbs.

 Point 9: Conclusion: line 168 delete rationalize; line 169 studied approaches to study.

 Response 9: We have deleted rationalize and replaced studied approaches to study.

Point 10: Conclusions: line 168 delete rationalize; studies approaches to study.

 Response 10: We have deleted rationalize and replaced studied approaches to study.

Reviewer 2 Report

This paper provides a systematic description of diverse cultural traits that never noticed as important. but in this paper, the cultural conditions and media is not descripted in details. It need to add this contents in this paper.

Author Response

Point: This paper provides a systematic description of diverse cultural traits that never noticed as important. but in this paper, the cultural conditions and media is not descripted in details. It need to add this contents in this paper.

Response: We have expanded the description of the cultural traits shown in the images.